# ESS-Flow: Training-free guidance of flow-based models as inference in source space

## Abstract

Guiding pretrained flow-based generative models for conditional generation or to produce samples with desired target properties enables solving diverse tasks without retraining on paired data. We present ESS-Flow, a gradient-free method that leverages the typically Gaussian prior of the source distribution in flow-based models to perform Bayesian inference directly in the source space using Elliptical Slice Sampling. ESS-Flow only requires forward passes through the generative model and observation process, no gradient or Jacobian computations, and is applicable even when gradients are unreliable or unavailable, such as with simulation-based observations or quantization in the generation or observation process. We demonstrate its effectiveness on designing materials with desired target properties and predicting protein structures from sparse inter-residue distance measurements.

## 1 Introduction

In generative modeling, we are given data samples and aim to construct a sampler that approximates the data distribution. Diffusion models (Ho et al., 2020; Song et al., 2021) and continuous normalizing flows (Lipman et al., 2023; Liu et al., 2023; Albergo et al., 2023) achieve this by transporting samples from a simple source distribution to the data distribution. Instead of just unconditional sampling, we often need to generate samples with specific properties or to match observed data in the context of inverse problems. If sufficient paired data is available, we can train conditional generative models (Dhariwal & Nichol, 2021; Rombach et al., 2022; Miller et al., 2024). However, this requires specialized models for each task and large amounts of paired training data.

Training-free conditional generation methods offer a more flexible alternative by reusing pretrained models. Instead of paired training data, these methods often use a likelihood function to measure how well a sample matches the desired observation. Guidance-based methods modify the transport map towards the target distribution using the conditional likelihood score (Song et al., 2021), though the score often has to be approximated. Alternatively, optimization-based methods minimize the negative log-likelihood with gradient descent, using the generative model as either an explicit regularizer through noising-denoising procedures (Martin et al., 2025; Levy et al., 2024) or an implicit one that constraints the gradient flow to the data manifold (Ben-Hamu et al., 2024).

While optimization-based methods are empirically shown to perform well in image inverse tasks, they only provide point estimates rather than samples, offer little guarantees against local optima, and fail when gradients are unreliable or unavailable. Notably, this occurs when the generative process involves non-differentiable operations like quantization, or when likelihood evaluations require non-differentiable simulations which is common in scientific applications (e.g. Alhossary et al., 2015; Alford et al., 2017). Some of the former limitations can be addressed by formulating controlled generation as Bayesian inference, where the pretrained generative model provides the prior, enabling inference methods that sample from the target distribution. However, many state-of-the-art methods for solving Bayesian inverse problems with generative model priors still requires gradients (Chung et al., 2023; Zhang et al., 2025; Janati et al., 2024; Wu et al., 2023) or are limited to linear Gaussian observations (Kelvinius et al., 2025; Cardoso et al., 2024), limiting their applicability.

Our key insight is that many pretrained diffusion and flow-based models can be recast as continuous normalizing flows with Gaussian source distributions. This enables us to perform Bayesian inference in the source space, instead of the complex data space, and use tailored Markov chain Monte Carlo

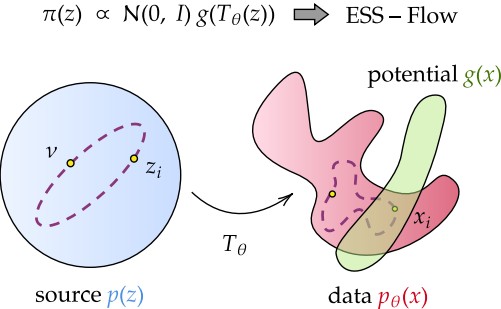

Figure 1: Illustration of ESS-Flow. We sample the target distribution $\pi(z)$ in the source space of flow-based models, which typically has a Gaussian prior $p(z)$. This allows gradient-free sampling with elliptical slice sampling, which defines an ellipse through the current MCMC iterate $z_i$ and a sample from the prior $\nu$, and moves to the next iterate by searching along the ellipse. At stationarity, the transformed ellipse passes through regions of high potential in the data space by construction.

(MCMC) methods like elliptical slice sampling (Murray et al., 2010). This results in ESS-Flow, a new training-free controlled generation method, approximating the target distribution in the source space of flow-based models, as illustrated in Figure 1.

ESS-Flow is by design gradient-free, requiring only point-wise evaluations of the generative model and the likelihood (or more generally, potential) function. This makes it particularly valuable for problems involving quantization, such as molecular and material design with categorical atomic numbers, where gradients are not well-defined. It is applicable for arbitrary, possibly non-linear and non-differentiable, potential functions and does not require knowledge of the noising process used during model training, only the final trained transport map. Furthermore, contrary to guidance-based methods, ESS-Flow preserves the pretrained velocity field. As discussed by Wang et al. (2025), this has the benefit that if the prior vector field is optimized for fast generation, for example when trained with minibatch-OT coupling (Pooladian et al., 2023; Tong et al., 2024), this property is retained by source sampling methods like ESS-Flow. ESS-Flow also extends naturally to flows on manifolds, such as those trained with Riemannian flow matching, as long as the source distribution is Gaussian.

We summarize our main contributions as:

- We identify and illustrate limitations of gradient-based methods for controlled generation.
- We present ESS-Flow, a training-free and asymptotically exact sampling method for flow-based models, which requires no gradients through the generative model or the potential function.
- We propose a multi-fidelity extension of ESS-Flow, leveraging the fact that flow-based generative models in practice are simulated from using a numerical solver, to improve the computational efficiency of the method.
- We demonstrate ESS-Flow on materials design with target properties and protein structure prediction from partial inter-residue distance measurements, achieving lower mean absolute errors on materials and improved structural realism in proteins.

While the gradient-free nature of ESS-Flow is highly beneficial in many settings, it also limits the applicability of the method in situations when the prior poorly informs the target distribution, for instance when the target is constrained on a lower-dimensional manifold. The primary use-case for ESS-Flow is thus applications, e.g. in scientific domains, where the target distribution is not overly-collapsed. We discuss this limitation further below.

## 2 PRELIMINARIES

**Flow-based generative models**: Consider a generative model that transports samples $x_0$ from a simple source distribution at $t = 0$ to samples $x_1$ from the data distribution at $t = 1$ through a

learned velocity field $u_t^\theta(x)$ by solving the ordinary differential equation (ODE):

$$x_1 = T_\theta(x_0) \coloneqq x_0 + \int_0^1 u_t^\theta(x_t)\, \mathrm{d}t.\tag{1}$$

Both diffusion models with the probability flow ODE for generation (Song et al., 2021) and models trained with the conditional flow matching objective (Lipman et al., 2023; Albergo et al., 2023) can be viewed as instances of this model class. We collectively refer to them as flow-based generative models with a transport map $T_\theta$ and consider the case where the source distribution is Gaussian, i.e., the model defines a continuous normalizing flow. To simplify notation, we drop the time subscript $t$ and denote source samples $x_0$ by $z$ and data samples $x_1$ by $x$, i.e., the generative model prior is defined by $x = T_\theta(z)$ with $z \sim \mathcal{N}(0, I)$.

**Controlled generation as Bayesian inference**: In controlled generation tasks, we seek to sample from a target distribution $\pi(x) \propto g(x)p_\theta(x)$, where it is assumed that the normalization constant is finite so that $\pi(x)$ is indeed a probability distribution. Here, $p_\theta(x)$ is the prior (unconditional) generative model and $g(x)$ is a nonnegative potential function that guides samples toward desired properties. Under this formulation, $\pi(x)$ is the posterior distribution $p(x|y)$ when $g(x)$ is a likelihood function $p(y|x)$ conditioned on observation $y$, or a tilted prior when $g(x)$ is a general reward function. In our setting, the prior data distribution is given by the pretrained flow-based generative model of the form in equation (1).

## 3 RELATED WORK

Many methods have been proposed for controlled generation with diffusion and flow-based models (see surveys in, e.g., Daras et al., 2024; Chung et al., 2025; Zhao et al., 2025). We focus on those that are training-free and applicable to ODE-based generative models of the form in equation (1).

Methods such as DPS (Chung et al., 2023), FlowDPS (Kim et al., 2025), and OT-ODE (Pokle et al., 2024) incorporate a guidance term into the transport map, obtaining target samples in a single pass through the modified generative process. This term involves the score of the conditional likelihood, which is often approximated, and the sequential process cannot correct for errors that occur early in the generation. Methods like PnP-Flow (Martin et al., 2025), DAPS (Zhang et al., 2025), and DDSMC (Kelvinius et al., 2025) overcome this limitation by alternating between noise space and the data space with progressively annealed noise. PnP-Flow optimizes in the data space and regularizes through noising-denoising steps between gradient updates. DAPS generalizes this framework for posterior sampling and DDSMC extends DAPS using sequential Monte Carlo for linear Gaussian potentials. These methods do not require differentiating the transport map and perform approximate posterior updates in the data space at each iteration of the generative procedure. However, they require access to the noising process used during training, as the noising step is part to the algorithm.

Source space methods like D-Flow (Ben-Hamu et al., 2024) performs gradient-based source point optimization, relying on implicit regularization that leads to manifold-constrained gradient flow in the data space. Purohit et al. (2025) extend D-Flow to posterior sampling in the source using Langevin Monte Carlo. In work which is concurrent to ours, Wang et al. (2025) use Hamiltonian Monte Carlo in the source space, which has also been considered earlier by Graham & Storkey (2017) who demonstrated latent space sampling with constrained Hamiltonian Monte Carlo for variational autoencoders. A key limitation of existing source space methods is their reliance on gradients, requiring expensive backpropagation through the ODE solver. To the best of our knowledge, our method, ESS-Flow is the only gradient-free method in this category. Furthermore, it does not require knowledge of the noising process used in training, only the trained transport map.

## 4 METHOD

Suppose that we have a generative model prior $p_\theta(x)$ from a flow-based model that maps samples from a Gaussian source distribution $z \sim \mathcal{N}(0, I)$ to the data space with the transport map $x = T_\theta(z)$. Given a potential function $g(x)$, such as the likelihood $p(y|x)$ for an observation $y$ or a general reward function, we seek samples from the target $\pi(x)$:

$$\pi(x) \propto g(x)\, p_\theta(x) = g(x)\, p(z)\, |\det(J_{T_\theta}(z))|^{-1},\tag{2}$$

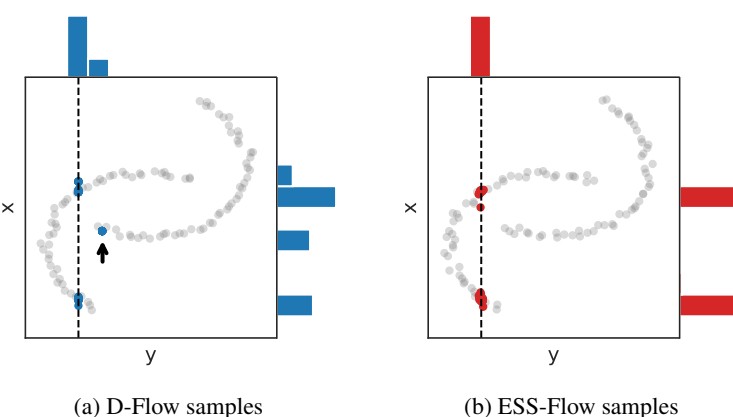

(a) D-Flow samples       (b) ESS-Flow samples

Figure 2: Conditional generation targeting a specific $y$ indicated by the dotted line. Prior samples are shown in gray with marginals along the border. Some D-Flow samples, which follow the source-space gradient, become trapped in disconnected manifolds (indicated by arrow), while ESS-Flow, being gradient-free, can avoid this problem.

where $z = T_\theta^{-1}(x)$ and $J_{T_\theta}(z)$ denotes the Jacobian of the transport map evaluated at $z$.

### 4.1 ESS-FLOW: AVOIDING GRADIENT COMPUTATIONS FOR SOURCE SPACE SAMPLING

Sampling the target distribution $\pi(x)$ or optimizing for a point estimate in the data space requires evaluating the Jacobian of the transport map, which is computationally expensive for continuous flow-based models. Methods like PnP-Flow (Martin et al., 2025) avoid this by maximizing only $g(x)$ and regularizing with an iterative noising-denoising procedure, which does not guarantee convergence to the maximum a posteriori (MAP), nor sampling from the target $\pi(x)$.

With a change of variables, we can instead approximate the target distribution $\pi(z)$ in the source:

$$\pi(z) = \pi(x)\,|\det(J_{T_\theta}(z))| \propto g(x)\,p(z)\,|\det(J_{T_\theta}(z))|^{-1}\,|\det(J_{T_\theta}(z))| = g(T_\theta(z))\,p(z). \quad (3)$$

Note the cancellation of the Jacobian terms due to the fact that we express both the prior and the posterior in the source space. While we no longer need the Jacobian for point-wise density evaluation, optimizing for a MAP estimate or using gradient-based sampling with Langevin Monte Carlo (Purohit et al., 2025) or Hamiltonian Monte Carlo (Graham & Storkey, 2017; Wang et al., 2025) still requires it, since the gradient $\nabla_z g(T_\theta(z)) = J_{T_\theta}(z)^\top \nabla_x g(x)$ involves the Jacobian. Beyond computational expense, gradients may be entirely unavailable when the generative process involves quantization, or when evaluating the potential requires non-differentiable simulators or external programs that are not amendable to automatic differentiation as in Section 5.1.

Furthermore, gradient-based methods can struggle when the prior has multiple disconnected modes. To illustrate this, consider D-Flow applied to the toy-problem in Figure 2, where $p_\theta(x)$ consists of two interleaving half-circles and we seek samples with a specific target $y$. D-Flow maximizes $g(T_\theta(z))$ w.r.t. $z$ using gradient-based optimization while relying on implicit regularization. A small gradient step $\delta_z \propto J_{T_\theta}(z)^\top \nabla_x g(x)$ in the source space corresponds to the step $\delta_x = J_{T_\theta}(z)J_{T_\theta}(z)^\top \nabla_x g(x)$ in the data space. This projects gradients along the data manifold, resulting in a manifold-constrained gradient flow (Ben-Hamu et al., 2024). However, as seen in Figure 2, even with a large learning rate that deviates significantly from a gradient flow, many D-Flow samples remain trapped within the disconnected manifold component where they are initialized.

To overcome the limitations of gradient-based methods, we propose to use elliptical slice sampling (ESS, Murray et al., 2010) to sample from the target distribution in the source space. This leverages the fact that the prior $p(z)$ is a multivariate Gaussian, whereas the complexity of the transport map $T_\theta(z)$ has been incorporated in the potential according to equation (3). This is precisely the setup where ESS excels. Our resulting approach, ESS-Flow, requires only point-wise evaluations of the generative model and involves no gradient or Jacobian computation. Being an MCMC method, ESS-Flow is flexible with initialization, e.g., by sampling $z_0$ from the Gaussian prior, and

then proceeding iteratively. As illustrated in Figure 1, a proposal is drawn from an ellipse in the source space defined by the current state $z_i$ and a random sample $\nu$ from the Gaussian prior. Assuming that the transport map $T_\theta$ is continuous in $z$, it nonlinearly maps the ellipse to a continuous, connected curve in the data space that passes through the current state $x_i$. The potential function $g(T_\theta(z))$ evaluated in the data space determines whether the proposal will be accepted. Unlike Metropolis-Hastings, where a rejected proposal requires drawing a new proposal from a new ellipse, ESS adaptively shrinks the angular bracket around the current ellipse when a proposal is rejected, excluding regions that contain the rejected proposal while maintaining support near the current state, and draws a new proposal from the shrunken ellipse. This mechanism can be seen as an adaptive step size that enables larger jumps without tuning. As discussed by Murray et al. (2010), this guarantees that the procedure will terminate by accepting a proposal in finite time when the pullback potential function $g \circ T_\theta$ is continuous. However, it excludes potentials that constrain the target distribution to a lower-dimensional manifold in the source space, such as exact equality constraints. Searching along the ellipse in the source corresponds to gradient-free exploration of the target data space, and the resulting Markov chain in the source $\{z_i\}$ yields the corresponding Markov chain $\{x_i\}$ when passed through the transport map. ESS-Flow has minimal hyperparameters to tune, and one MCMC step is summarized in Algorithm 1.

The convergence of ESS is discussed by Murray et al. (2010) who show that the induced Markov kernel leaves the target distribution invariant. A more detailed convergence analysis is provided by Hasenpflug et al. (2025) and Natarovskii et al. (2021). For completeness we state one of the main results here, adapted to our setting of flow-based models. We also present numerical evaluations for the scaling of ESS-Flow with dimensions in Appendix A.1.

**Proposition 1 (Geometric convergence of ESS-Flow, Theorem 2.2 by Natarovskii et al. (2021))**
*Suppose that the pullback potential $z \mapsto g \circ T_\theta(z)$ is bounded away from 0 and $\infty$ on compact sets and has regular tail behavior (details in Natarovskii et al. (2021), Assumption 2.1), then the ESS Markov chain, which we denote $\nu(\cdot, x)$, converges geometrically fast to the target measure $\pi$:*

$$\|\nu^n(\cdot, z) - \pi\|_{\mathrm{TV}} \le c\,(1 + \|z\|_2)\,\beta^n, \quad \forall z \in \mathbb{R}^d,$$

*where $\|\cdot\|_{\mathrm{TV}}$ stands for the total variation distance, $\nu^n$ is the $n$-th iteration of the chain, and $c > 0$ and $0 < \beta < 1$ are constants.*

---

**Algorithm 1** ESS-Flow: one MCMC iteration

---

**Require:** source $\mathcal{N}(0, I)$, transport map $T_\theta$, potential $g$, current state-potential $(z, g(T_\theta(z)))$

---

1: Sample $\nu \sim \mathcal{N}(0, I)$, $u \sim U(0, 1)$
2: Sample $\theta \sim U(0, 2\pi)$
3: Initialize bracket $[\theta_l, \theta_u] \leftarrow [\theta - 2\pi, \theta]$
4: **while** true **do**
5:     $z' = z \cos\theta + \nu \sin\theta$
6:     $x' = T_\theta(z')$
7:     **if** $\log g(x') > \log g(x) + \log u$ **then**
8:        **return** $(z', g(x'))$
9:     **else**
10:        $\theta_l, \theta_u \leftarrow \textsc{ShrinkBracket}(\theta, \theta_l, \theta_u)$
11:        Resample $\theta \sim U(\theta_l, \theta_u)$

12: **function** $\textsc{ShrinkBracket}(\theta, \theta_l, \theta_u)$
13:     **if** $\theta < 0$ **then**
14:        $\theta_l \leftarrow \theta$
15:     **else**
16:        $\theta_u \leftarrow \theta$
17:     **return** $\theta_l, \theta_u$

---

## 4.2 Multi-fidelity sampling with ESS-Flow

Flow-based generative models are defined in continuous time according to equation (1), but in practice the transport map is solved numerically with finite discretization $\Delta$ rather than exactly. This means that we in principle have access to a class of approximate priors $\{p_\theta^\Delta(x) : \Delta \in (0, 1)\}$ corresponding to different discretization levels, where, intuitively, smaller $\Delta$ gives rise to more accurate models.

Our proposed MCMC based sampler could be generalized in different ways to take advantage of such a multi-fidelity setup. The high-level idea is to rely on coarser, and thus computationally

cheaper, evaluations of the transport map for a large portion of the evaluations, while ensuring that the final samples nevertheless target a prespecified high-fidelity model. This could be accomplished by delayed acceptance ESS (Bitterlich et al., 2025), or by methods resembling parallel tempering (Earl & Deem, 2005) or simulated tempering (Marinari & Parisi, 1992) where the annealing temperature is replaced by the discretization level. A simpler approach, which we elaborate here as a proof of concept, is a post-correction of the generated samples using importance weighting.

Let $T_\theta^\Delta(z)$ denote the transport map with coarse discretization $\Delta$ and $T_\theta^\delta(z)$ with fine discretization $\delta \ll \Delta$. The target distribution from equation (3) for the high-fidelity model can be rewritten as:

$$\pi^\delta(z) \propto g(T_\theta^\delta(z)) \, p(z) = \frac{g(T_\theta^\delta(z))}{g(T_\theta^\Delta(z))} g(T_\theta^\Delta(z)) \, p(z) \propto w \, \pi^\Delta(z) \tag{4}$$

We sample from ESS-Flow targeting $\pi^\Delta(z)$ using the coarse transport map $T_\theta^\Delta$ and re-weight samples with self-normalized importance weights $w_i \propto g(T_\theta^\delta(z_i))/g(T_\theta^\Delta(z_i))$. The expensive high-fidelity transport map is evaluated only on the final MCMC samples, not during the ESS itself, significantly reducing computational cost while maintaining accuracy of the target.

## 5 EXPERIMENTS

We evaluate ESS-Flow on generating materials with target properties and predicting protein backbone structures from sparse inter-residue distances. We compare against state-of-the-art optimization-based methods: D-Flow (Ben-Hamu et al., 2024), PnP-Flow (Martin et al., 2025) and ADP-3D (Levy et al., 2024); and sampling-based method: DAPS (Zhang et al., 2025).

### 5.1 MATERIAL GENERATION

We use FlowMM (Miller et al., 2024), a flow-based generative model trained with Riemannian flow matching on the MP-20 dataset (Jain et al., 2013), as the prior over materials. A crystalline material $c$ with unit lattice consisting of $n$ atoms is represented by the tuple $(\boldsymbol{a}, \boldsymbol{f}, \boldsymbol{l}, \boldsymbol{\beta})$, where $\boldsymbol{a} \in \{-1, 1\}^{n \times 7}$ are the binarized atomic numbers, $\boldsymbol{f} \in [0, 1]^{n \times 3}$ are the fractional coordinates of each atom, $\boldsymbol{l} \in \mathbb{R}_+^3$ are the lattice cell lengths, and $\boldsymbol{\beta} \in [60°, 120°]^3$ are the lattice angles. We convert the uniform and log-normal source distributions of $\boldsymbol{f}, \boldsymbol{l}, \boldsymbol{\beta}$ into a standard Gaussian via a change of variables.

To enable comparison with gradient-based methods, we predict material properties using auto-differentiable ALIGNN (Choudhary & DeCost, 2021) models trained on the JARVIS-DFT (Choudhary et al., 2020) dataset, rather than simulation-based procedures. For target properties, we consider bulk modulus, shear modulus, and band gap, choosing target values above the 99th percentile of the prior property distribution. We also evaluate guiding generation toward stable materials, where stability is measured by the predicted energy above hull. To demonstrate ESS-Flow's performance in a truly non-differentiable setting, we target materials with specific space-group symmetry. For this task, the potential function is a binary indicator computed using a non-differentiable external program (Togo et al., 2024), making gradient-based methods inapplicable. Since FlowMM requires specifying the number of atoms per unit lattice, we choose values based on the distribution of atoms for materials with large property values. Target property values, number of atomic sites per lattice, and potential function details are summarized in Table 1.

Table 1: Target properties and values for the material generation experiments

| Property $P_c$ | Target $y$ | Atoms $n$ | Potential function $g(c)$ | Scale $\sigma_y$ |
|---|---|---|---|---|
| Bulk modulus | 300 GPa | 4 | $\exp(-(P_c - y)^2/2\sigma_y^2)$ | 10 GPa |
| Shear modulus | 200 GPa | 4 | $\exp(-(P_c - y)^2/2\sigma_y^2)$ | 10 GPa |
| Band gap | 10 eV | 12 | $\exp(-(P_c - y)^2/2\sigma_y^2)$ | 0.1 eV |
| Energy above hull | - | 8 | $\exp(-P_c/\sigma_y)$ | 0.01 eV |
| Space group | P6$_3$/mmc | 8 | $\mathbf{1}[P_c = y]$ | - |

We apply dimension-wise learning rates for all methods, as gradient magnitudes vary between $(\boldsymbol{a}, \boldsymbol{f}, \boldsymbol{l}, \boldsymbol{\beta})$. FlowMM outputs $\tilde{\boldsymbol{a}} \in \mathbb{R}^{n \times 7}$, a soft 7-bit encoding that gets rounded to discrete values

Table 2: Mean and standard deviation (in parentheses) of absolute errors between the sample properties and targets, and mean and standard deviation of energy above hull values.

| Method | Bulk modulus | Shear modulus | Band gap | Energy above hull |
|---|---|---|---|---|
| Unconditional | 209.39 (59.07) | 168.41 (25.95) | 9.28 (1.15) | 1.96 (1.34) |
| D-Flow | 205.88 (60.79) | 165.93 (27.72) | 9.24 (1.18) | 1.92 (1.38) |
| PnP-Flow | 49.93 (32.33) | 75.48 (33.45) | 5.63 (1.03) | -0.02 (0.06) |
| DAPS | 39.14 (26.47) | 84.33 (37.10) | 3.90 (1.67) | -0.06 (0.05) |
| ESS-Flow | **8.99 (6.69)** | **10.53 (9.21)** | **1.85 (1.66)** | **-0.19 (0.07)** |

$a$. ALIGNN expects integer atomic numbers $k_i$ for the $i$-th atom to create atom embeddings $\boldsymbol{h}_i$ by selecting the $k_i$-th row of its embedding matrix $\mathbf{E}$. To maintain differentiability for D-Flow and PnP-Flow, we use the continuous values $\tilde{k}_i$, derived from $\tilde{a}_i$, to create soft atom embeddings $\boldsymbol{h}_i$:

$$z_{ij} = (j - \tilde{k}_i)^2, \quad \boldsymbol{v}_i = \text{softmax}(-\boldsymbol{z}_i/\tau), \quad \boldsymbol{h}_i = \mathbf{E}^\mathsf{T}\boldsymbol{v}_i, \tag{5}$$

where we set $\tau = 0.1$. For DAPS, we avoid this approximation by using Langevin Monte Carlo only for $(\boldsymbol{f}, \boldsymbol{l}, \boldsymbol{\beta})$ and sampling $\boldsymbol{a}$ with Metropolis–Hastings using proposals that flip one bit of the binarized atomic numbers. The noising and denoising steps are adapted from the original diffusion formulation to match FlowMM. Hyperparameter details and the runtime costs of the methods are provided in the Appendix.

We measure performance using the absolute errors between sample properties and targets for conditional generation, and the predicted energy above hull for stability, as shown in Table 2. Even with the continuous approximation for $\boldsymbol{a}$, D-Flow fails to explore atomic compositions far from initialization. PnP-Flow, which optimizes in data space, is less restricted by this limitation. By avoiding gradient-based exploration of $\boldsymbol{a}$, DAPS and ESS-Flow perform better; specifically, ESS-Flow outperforms all other methods significantly with the lowest errors. The approximation quality is further illustrated in Figure 3, where we see that ESS-Flow better recovers the sharp target distributions. ESS-Flow also successfully generates 92.3% of samples with the target P6$_3$/mmc space group, compared to only 2.5% when sampling unconditionally from the prior.

We assess the quality of generated samples by computing the S.U.N.T. rate (stability, uniqueness, novelty and threshold). Valid materials have lattice volume greater than 0.1 Å$^3$, atomic numbers within range, and balanced charge. After filtering invalid materials, we follow FlowMM and relax lattice structures with CHGNet (Deng et al., 2023), without further density functional theory based relaxations. Stability rate (S.) is the proportion of relaxed structures with energy above hull $<0.1$ eV/atom according to CHGNet. This procedure differs from the predictor used to guide generation of meta-stable materials and can thus produce different results. Uniqueness and novelty rates (U.N.) measure the proportion of unique samples among themselves and with respect to the training set, respectively, using pymatgen's `StructureMatcher` (Ong et al., 2013) with default settings. The proportion of samples meeting the desired property (T.) is measured as those within 5% of the target property for bulk modulus, shear modulus and band gap, less than 0 eV for stable materials, or those with space group as P6$_3$/mmc depending on the task. The combined S.U.N.T. rates computed over 1000 generated samples are reported in Table 3.

ESS-Flow successfully targets the desired properties and achieves the highest S.U.N.T. rates across all tasks. The S.U.N. rates are naturally low compared to unconditional generation, but they should be viewed in light of the fact that we are (successfully) targeting extreme values for the desired properties, with target values set to the 99th percentile based on the unconditional model. Despite this challenging controlled generation problem, the resulting rates indicate that ESS-Flow does generate a subset of materials that are good candidates for further exploration.

### 5.1.1 EVALUATION OF MULTI-FIDELITY SAMPLING

We perform preliminary evaluation of the suggested proof of concept for multi-fidelity sampling in Section 4.2. ESS-Flow samples obtained by evaluating the potential with a coarse discretization $\Delta = 1/50$ are re-weighted with importance weights computed using a much finer discretization $\delta = 1/1000$ for the ODE, as described in equation (4). The weighted samples have reasonable effective sample sizes of 65.3% and 33.9% for the bulk modulus and shear modulus tasks, respectively.

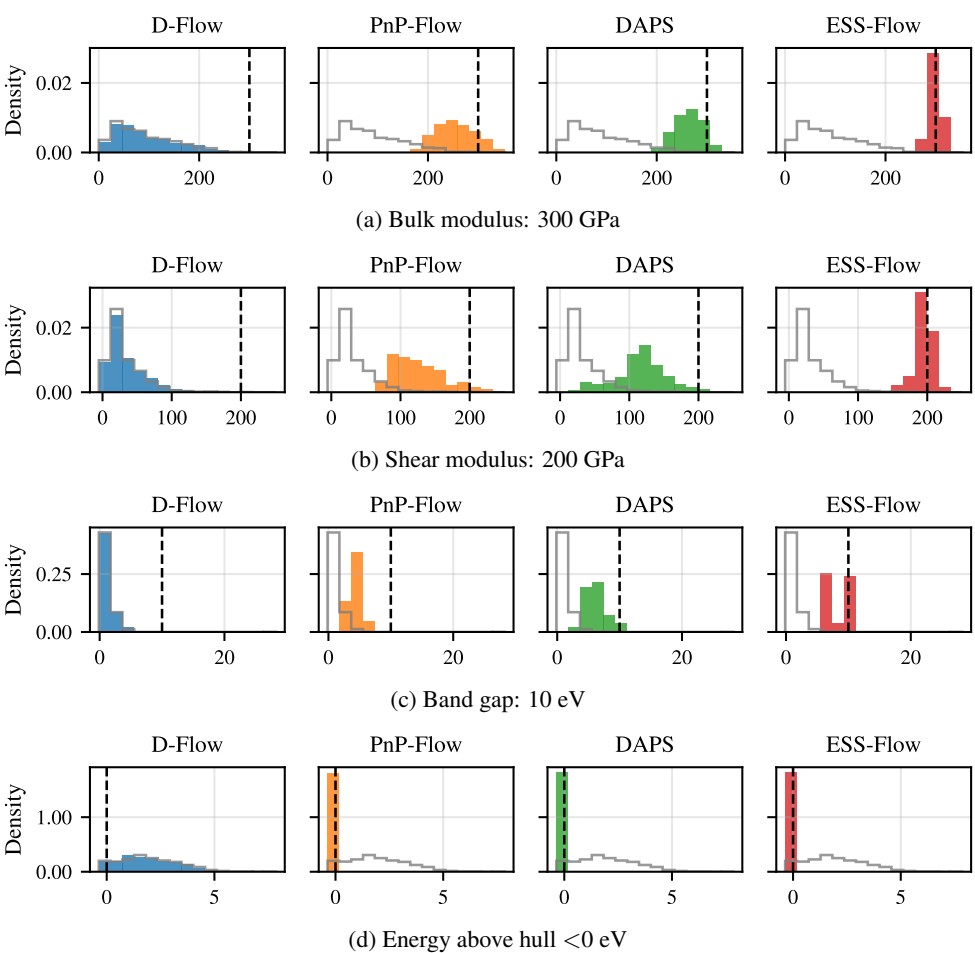

(a) Bulk modulus: 300 GPa

(b) Shear modulus: 200 GPa

(c) Band gap: 10 eV

(d) Energy above hull $<0$ eV

Figure 3: Sample property distributions. Property values are along the X-axis, prior property distribution is shown in gray, and target values are shown as dotted lines. ESS-Flow samples are near target values across all tasks.

However, for the band gap and stability tasks, which have sharper target distributions, the effective sample sizes are much lower, 0.1% and 1.0%, respectively. This is a shortcoming of the simple importance re-weighting approach, which assigns disproportionately large weights to samples with low potentials under the coarse discretization. Further analysis comparing ESS-Flow samples and the weighted samples is provided in Appendix A.3.

## 5.2 PROTEIN STRUCTURE PREDICTION

For protein backbone structure prediction from partial inter-residue distances, we follow Levy et al. (2024). We use Chroma (Ingraham et al., 2023), a pretrained diffusion model, as our prior over protein backbone structures denoted by $x \in \mathbb{R}^{4n \times 3}$, which are the 3D coordinates of all four heavy atoms from $n$ amino acid residues. Since we require a deterministic mapping between source and data samples, we modify Chroma's random protein graph construction to use k-nearest neighbors and generate samples with the probability flow ODE.

Levy et al. (2024) compute all pairwise distances between $\alpha$-carbons in protein PDB:7r5b (Warstat et al., 2023) containing 147 residues and randomly sample distances without noise. However, they note that this is simplified, as nuclear magnetic resonance spectroscopy cannot probe distances above 6 Å. We account for this by sampling only 300 out of 330 distances that are $<6$ Å and adding Gaussian noise $\mathcal{N}(0, 0.5^2)$, as the protein is deposited at the resolution of 1.77 Å. This results in a highly underdetermined inverse problem, suggesting that a Bayesian approach is suitable for captur-

Table 3: Quality of ESS-Flow samples obtained in different tasks.

| Task | Method | Valid | S. | U.N. | T. | S.U.N. | S.U.N.T. |
|---|---|---|---|---|---|---|---|
| Bulk modulus | Unconditional | 77.9 | 32.4 | 73.2 | 0.4 | 28.3 | 0.1 |
| | D-Flow | 75.2 | 31.0 | 70.2 | 0.4 | 26.3 | 0.0 |
| | PnP-Flow | 94.8 | 46.2 | 71.4 | 17.8 | 29.0 | 3.8 |
| | DAPS | 95.4 | 57.6 | 80.8 | 19.8 | 47.0 | 9.4 |
| | ESS-Flow | 98.3 | 56.4 | 46.1 | 79.6 | 19.3 | **13.7** |
| Shear modulus | Unconditional | 76.6 | 31.4 | 71.9 | 0.0 | 27.3 | 0.0 |
| | D-Flow | 78.6 | 32.2 | 73.2 | 0.0 | 27.5 | 0.0 |
| | PnP-Flow | 92.7 | 43.2 | 68.3 | 5.4 | 25.7 | 0.9 |
| | DAPS | 86.2 | 47.2 | 74.6 | 1.8 | 39.1 | 0.5 |
| | ESS-Flow | 98.2 | 50.3 | 30.5 | 59.9 | 10.0 | **4.7** |
| Band gap | Unconditional | 68.0 | 14.5 | 68.0 | 0.0 | 14.5 | 0.0 |
| | D-Flow | 69.8 | 16.2 | 69.7 | 0.0 | 16.1 | 0.0 |
| | PnP-Flow | 44.0 | 1.6 | 43.8 | 0.2 | 1.6 | 0.1 |
| | DAPS | 7.1 | 0.4 | 7.1 | 0.0 | 0.4 | 0.0 |
| | ESS-Flow | 48.8 | 16.9 | 48.0 | 23.2 | 16.5 | **16.0** |
| Energy above hull | Unconditional | 69.9 | 25.1 | 69.0 | 2.3 | 24.2 | 1.4 |
| | D-Flow | 69.7 | 25.2 | 68.8 | 2.7 | 24.3 | 1.8 |
| | PnP-Flow | 79.8 | 52.6 | 73.9 | 57.4 | 47.2 | 34.5 |
| | DAPS | 47.0 | 19.9 | 46.8 | 42.0 | 19.7 | 17.6 |
| | ESS-Flow | 73.3 | 44.5 | 64.9 | 73.3 | 37.6 | **37.6** |
| Space group | Unconditional | 73.0 | 26.5 | 72.4 | 2.3 | 26.0 | 1.3 |
| | ESS-Flow | 87.1 | 50.0 | 54.6 | 81.9 | 26.7 | **25.5** |

ing the diversity of structures that agree with the observed data. Conditioned on these observations, we generate 10 backbone structures with D-Flow, ADP-3D, DAPS, and ESS-Flow. We report the $L^2$ distance $d_y$ between observations and corresponding sample distances, and the root mean square distance $\text{RMSD}_{\text{gt}}$ between the ground truth structure and samples in Table 4.

Table 4: Mean and standard deviation (in parenthesis) of the metrics for the sampled structures.

| Method | $d_y$ | $\text{RMSD}_{\text{gt}}$ | min. $\text{RMSD}_{\text{gt}}$ | ELBO | # Clashes |
|---|---|---|---|---|---|
| Unconditional | 80.21 (9.69) | 16.98 (0.83) | 15.53 | 8.70 (0.20) | 10.1 (7.5) |
| D-Flow | 46.54 (8.58) | 14.44 (1.35) | 11.54 | 8.64 (0.35) | 14.8 (13.5) |
| ADP-3D | 3.43 (0.34) | 11.45 (1.52) | 8.91 | -5.68 (1.24) | 731.3 (209.6) |
| DAPS | 11.79 (1.67) | 11.41 (1.17) | 9.47 | -8.07 (1.27) | 483.3 (114.0) |
| ESS-Flow | 37.02 (5.06) | 13.55 (1.32) | 10.63 | 8.89 (0.21) | 24.8 (17.1) |

Following Levy et al. (2024), we use the lower bound on the log marginal data likelihood (ELBO) from Chroma as a measure of structural realism. We also compute the clash count in the structures from each method as the number of instances where the van der Waals radii of atoms overlap. While $L^2$ distances suggest that ADP-3D and DAPS samples best match observations, the ELBO values and clash counts reveal that the resulting structures are highly unnatural. In both methods, as noise levels are annealed, regularization from the pretrained generative model diminishes, shifting optimization toward maximum likelihood estimates. This results in prioritizing good data fit (low RMSD) at the expense of weaker prior regularization, producing unrealistic structures. ESS-Flow explicitly enforces the prior, resulting in comparably more realistic samples. However, the high RMSD values indicate that this problem remains challenging for all methods we consider, including ESS-Flow, leaving room for improvement.

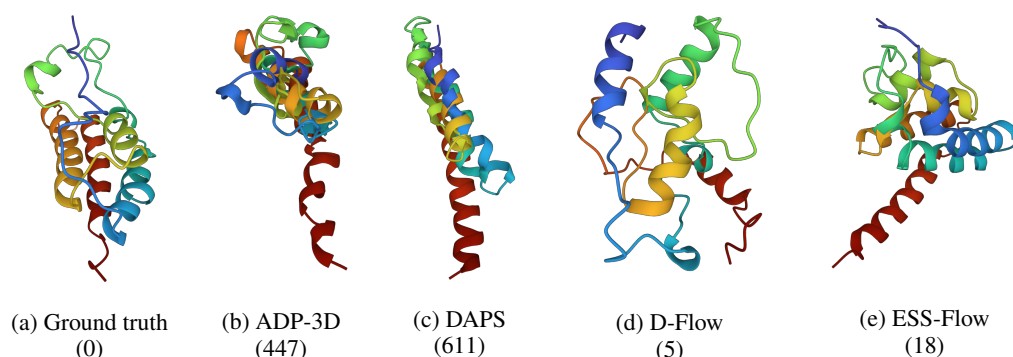

|              |              |           |              |               |
|:------------:|:------------:|:---------:|:------------:|:-------------:|
| (a) Ground truth | (b) ADP-3D | (c) DAPS | (d) D-Flow | (e) ESS-Flow |
| (0) | (447) | (611) | (5) | (18) |

Figure 4: Ground truth protein structure `PDB:7r5b`, conditional sample with the lowest $\text{RMSD}_{\text{gt}}$ from each method, and their clash counts (in parenthesis). While samples from ADP-3D and DAPS appear unnatural, ESS-Flow achieves a better trade-off between data fidelity and sample realism.

## 6 CONCLUSION

We introduce ESS-Flow, a training-free sampling method for controlled generation with pretrained flow-based generative models. The method requires minimal hyperparameter tuning and is entirely gradient-free, making it particularly valuable for problems where gradient-based optimization struggles, such as when the generative process involves quantization, or evaluating the potential requires non-differentiable simulations. However, this gradient-free approach limits ESS-Flow's effectiveness when the prior does not well inform the target distribution, such as in noiseless image inpainting tasks. Currently, we use moderate numbers of function evaluations in the ODE solver, fewer than what is typically used for unconditional generation. We propose and evaluate a multi-fidelity setup where samples obtained by using coarse ODE discretization are re-weighted with importance weights, as a proof of concept. Future work could explore more principled combinations of coarse and fine ODE discretization to better guide exploration of target distributions. Additionally, developing adaptive strategies for problems where the prior poorly covers the target distribution could expand the applicability of the method to a broader range of controlled generation tasks.

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

## A  APPENDIX

### A.1  GAUSSIAN MIXTURE MODEL

We numerically evaluate how ESS-Flow scales with dimension using simulated data. The prior distribution is a Gaussian mixture model consisting of 25 components, with means $\mu_{i,j} = (i, j, \ldots, i, j) \in \mathbb{R}^{d_x}$ for $(i, j) \in \{-2, -1, 0, 1, 2\}^2$ and covariances $(1/8)^2 I_{d_x}$. The unnormalized weights of the mixture components are independently drawn from a $\chi^2$ distribution. We consider a linear-Gaussian observation model $y = Ax + \sigma_y^2 \epsilon$ with $\epsilon \sim \mathcal{N}(0, I_{d_y})$, so the posterior is also a Gaussian mixture that can be computed exactly. The setup is similar to that of Cardoso et al. (2023), except with rescaled means and covariances while maintaining the relative separation between mixture components.

To avoid model approximation errors, instead of learning a transport map from a standard normal source distribution to the Gaussian mixture, we use the analytical expression of the ODE derived in Albergo et al. (2023) corresponding to the linear interpolant $x_t = (1-t)x_0 + tx_1$. For ESS-Flow, we run 1000 parallel MCMC chains for 501 steps and collect the last sample. The ODE is numerically integrated using the midpoint method with a step size of 0.05. We report the Sliced Wasserstein Distance (SWD) between ESS-Flow samples and exact posterior samples in Table 5. The average number of likelihood evaluations per MCMC step within ESS, denoted by $n_{\text{ellipse}}$, is presented as a measure of sampling efficiency.

Table 5: Mean Sliced Wasserstein Distance (with standard error) between exact posterior samples and ESS-Flow samples, and mean number of likelihood evaluations per MCMC step in ESS (with standard deviation in parentheses), computed over 10 random seeds.

| $d_y$ | $d_x$ | SWD | $n_{\text{ellipse}}$ |
|---|---|---|---|
| 1 | 8 | $0.037 \pm 0.010$ | 2.8 (1.3) |
| | 80 | $0.044 \pm 0.019$ | 2.8 (1.1) |
| | 800 | $0.064 \pm 0.025$ | 2.7 (0.5) |
| 2 | 8 | $0.034 \pm 0.014$ | 3.2 (0.7) |
| | 80 | $0.034 \pm 0.009$ | 2.8 (1.2) |
| | 800 | $0.037 \pm 0.016$ | 3.2 (1.6) |
| 4 | 8 | $0.028 \pm 0.008$ | 3.5 (1.2) |
| | 80 | $0.031 \pm 0.014$ | 3.0 (1.4) |
| | 800 | $0.038 \pm 0.024$ | 4.1 (2.4) |

With increasing dimension, the sampling efficiency of ESS-Flow remains stable and the SWD remains low. For a qualitative evaluation, we visualize the first two dimensions of the posterior for one random seed with $d_x = 800$ and $d_y = 1$ in Figure 5.

### A.2  RUNTIME COST ANALYSIS FOR THE MATERIAL GENERATION TASKS

In this section, we compare the computational and memory requirements of the different methods used in the material generation tasks. In addition to the number of function evaluations (NFE) of the generative model $T_\theta$, we also report the NFE of the potential function $g$ for the specific hyperparameters we use. Gradient-based methods additionally incur the cost of backward passes through either or both the generative model and potential, which must be taken into consideration.

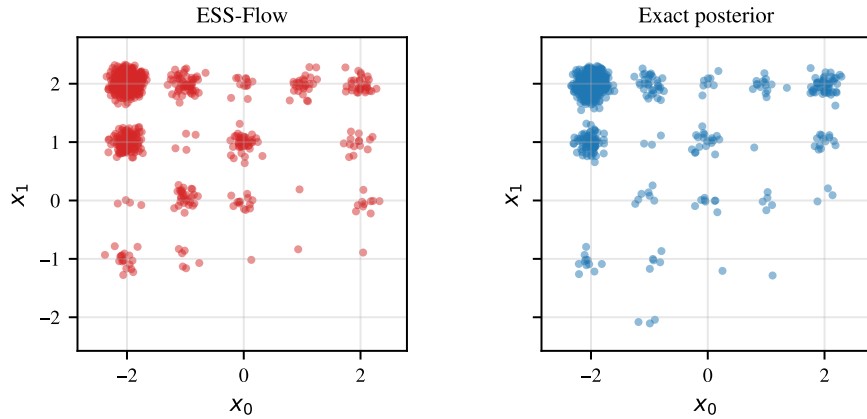

Figure 5: First two dimensions of the posterior approximated with ESS-Flow and the exact posterior for the GMM example with $d_x = 800$ and $d_y = 1$.

Unlike the other methods (Table 6), the NFE of ESS-Flow varies depending on the number of evaluations made along an ellipse per MCMC step ($n_{\text{ellipse}}$). We report the mean values for each task, computed over 1000 MCMC chains, in Table 7.

Table 6: Computational cost of the baselines used in material generation tasks

| Task | $\text{NFE}_{T_\theta}$ | $\text{NFE}_g$ | Gradients |
|---|---|---|---|
| Unconditional | $5 \times 10^1$ | $0$ | None |
| D-Flow | $5 \times 10^3$ | $1 \times 10^2$ | $T_\theta$ and $g$ |
| PnP-Flow | $5 \times 10^3$ | $1 \times 10^2$ | $g$ only |
| DAPS | $5 \times 10^2$ | $2 \times 10^3$ | $g$ only |

Table 7: Computational cost of ESS-Flow for the different material generation tasks

| Task | $n_{\text{ellipse}}$ | $\text{NFE}_{T_\theta}$ | $\text{NFE}_g$ | Gradients |
|---|---|---|---|---|
| Bulk modulus | 6.7 | $7 \times 10^4$ | $1 \times 10^3$ | None |
| Shear modulus | 7.1 | $7 \times 10^4$ | $1 \times 10^3$ | None |
| Band gap | 17.4 | $4 \times 10^5$ | $9 \times 10^3$ | None |
| Energy above hull | 12.0 | $1 \times 10^5$ | $2 \times 10^3$ | None |
| Space group | 4.8 | $5 \times 10^4$ | $1 \times 10^3$ | None |

The computational cost to generate one sample with ESS-Flow is significantly higher than the baselines. However, the overall cost can be reduced by using ESS-Flow with large batch sizes, as it has very low memory requirements. We report the peak GPU memory load for the different tasks and methods when using a batch size of 50 in Table 8.

### A.3 ADDITIONAL EVALUATION OF MULTI-FIDELITY SAMPLING

We evaluate the distribution of sample properties and quality of samples obtained by reweighting ESS-Flow samples with a high-fidelity transport map. We resample 1000 source samples according to their importance weights and obtain the corresponding material structures by integrating the ODE with discretization $\delta = 1/1000$. The distribution of sample properties is compared against that of the standard ESS-Flow samples in Figure 6, and performance metrics are reported in Table 9.

While the property distributions appear to better match the target posterior, the quality of the reweighted samples as measured by the S.U.N.T. rate in Table 10 is considerably worse, though metric

Table 8: Memory requirements of the methods in material generation tasks

| Task | Method | Peak GPU memory load (GB) |
|---|---|---|
| Bulk modulus | D-Flow | 6.91 |
| | PnP-Flow | 0.86 |
| | DAPS | 0.86 |
| | ESS-Flow | 0.29 |
| Shear modulus | D-Flow | 6.91 |
| | PnP-Flow | 0.86 |
| | DAPS | 0.86 |
| | ESS-Flow | 0.29 |
| Band gap | D-Flow | 62.22 |
| | PnP-Flow | 2.28 |
| | DAPS | 2.28 |
| | ESS-Flow | 0.6 |
| Energy above hull | D-Flow | 27.9 |
| | PnP-Flow | 1.57 |
| | DAPS | 1.57 |
| | ESS-Flow | 0.45 |
| Space group | ESS-Flow | 0.18 |

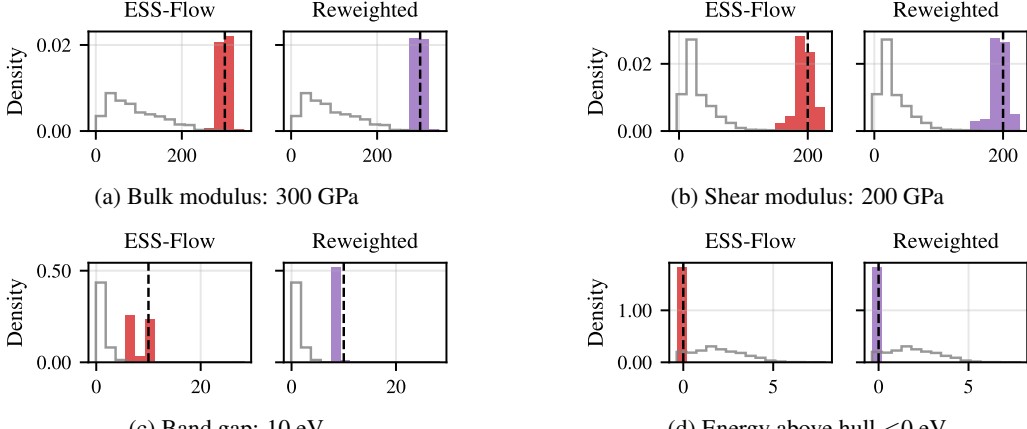

Figure 6: Sample property distributions between ESS-Flow and its importance reweighted version. Property values are along the X-axis, prior property distribution is shown in gray, and target values are shown as dotted lines.

Table 9: Mean and standard deviation (in parentheses) of absolute errors between the sample properties and targets, and mean and standard deviation of energy above hull values.

| Method | Bulk modulus | Shear modulus | Band gap | Energy above hull |
|---|---|---|---|---|
| ESS-Flow | 8.99 (6.69) | 10.53 (9.21) | 1.85 (1.66) | -0.19 (0.07) |
| Reweighted | 9.36 (6.70) | 10.00 (9.35) | 0.85 (0.47) | -0.22 (0.08) |

this may not capture all aspects of sample quality. The uniqueness rate drops due to resampling from the discrete weighted distribution, and this degradation is severe for the band gap task, where the effective sample size is only 0.1% as discussed in Section 5.1.1. This suggests that while the simple importance reweighting approach can improve accuracy for targets with moderate concentration, more sophisticated multi-fidelity approaches would be needed for sharply concentrated targets.

Table 10: Quality of ESS-Flow samples and its importance reweighted version for different tasks.

| Task | Method | Valid | S. | U.N. | T. | S.U.N. | S.U.N.T. |
|------|--------|-------|-----|------|-----|--------|----------|
| Bulk modulus | ESS-Flow | 98.3 | 56.4 | 46.1 | 79.6 | 19.3 | 13.7 |
| | Reweighted | 98.4 | 56.3 | 29.5 | 78.4 | 13.1 | 9.5 |
| Shear modulus | ESS-Flow | 98.2 | 50.3 | 30.5 | 59.9 | 10.0 | 4.7 |
| | Reweighted | 99.0 | 50.6 | 18.8 | 64.7 | 6.1 | 3.3 |
| Band gap | ESS-Flow | 48.8 | 16.9 | 48.0 | 23.2 | 16.5 | 16.0 |
| | Reweighted | 1.3 | 0.9 | 1.3 | 1.3 | 0.9 | 0.9 |
| Energy above hull | ESS-Flow | 73.3 | 44.5 | 64.9 | 73.3 | 37.6 | **37.6** |
| | Reweighted | 58.7 | 41.8 | 19.1 | 58.7 | 12.3 | 12.3 |

Table 11: Hyperparameters for the material generation tasks

| Method / Parameter | Bulk modulus | Shear modulus | Band gap | Energy above hull |
|--------------------|--------------|---------------|----------|-------------------|
| **Unconditional** | | | | |
| ODE integration steps $N$ | 50 | 50 | 50 | 50 |
| **D-Flow** | | | | |
| ODE integration steps $N$ | 50 | 50 | 50 | 50 |
| Optimizer | SGD | SGD | SGD | SGD |
| Optimization steps $N_{opt}$ | 100 | 100 | 100 | 100 |
| Learning rate $(\boldsymbol{a}, \boldsymbol{l}, \boldsymbol{\beta})$ | $10^{-6}$ | $10^{-5}$ | $10^{-4}$ | $10^{-4}$ |
| Learning rate $(\boldsymbol{f})$ | $10^{-8}$ | $10^{-7}$ | $10^{-6}$ | $10^{-6}$ |
| **PnP-Flow** | | | | |
| Optimizer | SGD | SGD | SGD | SGD |
| Optimization steps $N_{opt}$ | 100 | 100 | 100 | 100 |
| Learning rate decay | linear | linear | linear | linear |
| Min. learning rate | 0 | 0 | 0 | 0 |
| Max. learning rate $(\boldsymbol{a})$ | $10^{-4}$ | $10^{-3}$ | $10^{-1}$ | $10^{-2}$ |
| Max. learning rate $(\boldsymbol{f})$ | $10^{-6}$ | $10^{-5}$ | $10^{-3}$ | $10^{-4}$ |
| Max. learning rate $(\boldsymbol{l}, \boldsymbol{\beta})$ | $10^{-5}$ | $10^{-4}$ | $10^{-2}$ | $10^{-3}$ |
| ODE integration steps $N$ | $100 - n$ | $100 - n$ | $100 - n$ | $100 - n$ |
| **DAPS** | | | | |
| ODE integration steps $N$ | 5 | 5 | 5 | 5 |
| Optimization steps $N_{opt}$ | 100 | 100 | 100 | 100 |
| Sampling steps $N_s$ | 20 | 20 | 20 | 20 |
| Learning rate decay | linear | linear | linear | linear |
| Min. learning rate | $10^{-2}\alpha$ | $10^{-2}\alpha$ | $10^{-2}\alpha$ | $10^{-2}\alpha$ |
| Max. learning rate $(\alpha_{\boldsymbol{f}})$ | $10^{-3}$ | $10^{-3}$ | $10^{-5}$ | $10^{-4}$ |
| Max. learning rate $(\alpha_{\boldsymbol{l}}, \alpha_{\boldsymbol{\beta}})$ | $10^{-2}$ | $10^{-2}$ | $10^{-4}$ | $10^{-3}$ |
| $p(x_0|x_t)$ scale $(r_{\boldsymbol{a}})$ | $3(1-t)$ | $3(1-t)$ | $3(1-t)$ | $3(1-t)$ |
| $p(x_0|x_t)$ scale $(r_{\boldsymbol{f}}, r_{\boldsymbol{l}}, r_{\boldsymbol{\beta}})$ | $1-t$ | $1-t$ | $1-t$ | $1-t$ |
| **ESS-Flow** | | | | |
| ODE integration steps $N$ | 50 | 50 | 50 | 50 |
| MCMC steps | 201 | 501 | 201 | 201 |
| Burn-in | 200 | 500 | 200 | 200 |

## A.4 HYPERPARAMETER DETAILS

The hyperparameters for methods used in the material generation experiment and protein structure prediction are given in Table 11 and 12 respectively.

Table 12: Hyperparameters for the protein structure prediction

| Method / Parameter | |
|---|---|
| **Unconditional** | |
| ODE integration steps $N$ | 20 |
| **D-Flow** | |
| ODE integration steps $N$ | 20 |
| Optimizer | SGD |
| Optimization steps $N_{opt}$ | 500 |
| Learning rate schedule | `CyclicLR` |
| Min. learning rate | $10^{-6}$ |
| Max. learning rate | $10^{-5}$ |
| Step size up/down | 125 |
| Momentum | 0.9 |
| **ADP-3D** | |
| Optimizer | SGD |
| Optimization steps $N_{opt}$ | 1000 |
| Learning rate | 0.67 |
| Momentum | 0.99 |
| **DAPS** | |
| ODE integration steps $N$ | 5 |
| Optimization steps $N_{opt}$ | 1000 |
| Sampling steps $N_s$ | 5 |
| Learning rate decay | linear |
| Min. learning rate | $10^{-5}$ |
| Max. learning rate | $10^{-4}$ |
| $p(x_0|x_t)$ scale | $2t$ |
| **ESS-Flow** | |
| ODE integration steps $N$ | 20 |
| MCMC steps | 201 |
| Burn-in | 200 |

