# OpenReview forum: "ESS-Flow: Training-free guidance of flow-based models as inference in source space"
_ICLR.cc/2026/Conference — Submitted to ICLR 2026_

### Official Review · Reviewer_C4du · 2025-10-27

**Soundness:** 3
**Presentation:** 3
**Contribution:** 3
**Rating:** 6
**Confidence:** 4

**Summary:**

The authors propose ESS-Flow, a method to guide sampling from probability flow generative models without gradient-based guidance. Their method samples with MCMC along ellipsoids in the source space (Gaussian prior), and takes a step based on the potential /target property evaluated in the data space. They motivate their method with the limitations of gradient-based guidance and evaluate their guidance on protein and material design tasks.

**Strengths:**

- Guidance without gradients is a nice benefit of the proposed method
- The method is motivated well theoretically and with prior work
- The experiments suggest that ESS-Flow is able to effectively guide samples compared to gradient-based approaches

**Weaknesses:**

- Does guidance for certain properties improve the estimation of other properties/observables not used in guidance?
- Given that one of the benefits of the proposed method is not relying on gradients, it would strengthen the paper to show guidance for discrete target properties.
- The discussion of challenges about challenges with gradient-based guidance (Fig. 2) is interesting, and I think further discussion / experiments along these lines would further strengthen the paper. I wonder if there is a way to show something similar with more realistic data by artificially creating 2 disconnected modes (i.e. removing data in a transition region or something like that).
- "Limits ESS-Flow’s effectiveness when the prior does not well inform the target distribution": while I understand that things like image inpainting might be challenging for the method, have the authors evaluated ESS-Flow on other image guidance tasks?
- There are newer and more accurate models compared to something like CHGNet that might give more accurate metrics (MACE, eSEN, UMA, etc.)

**Questions:**

Please see above.

---

> ### Author Response · Authors · 2025-11-22
> **Response to Reviewer C4du**
>
> We thank the reviewer for their helpful feedback. Below we address the main concerns.
>
> ---
>
> ## 1. On properties not being guided
>
> Properties not included in the guidance potential will follow the distribution learned by the pretrained generative model, reflecting the correlations in the training data. If guidance on multiple properties is desired, ESS-Flow handles this naturally through a composite potential function $g(x) = \prod_{i} g_i(x)$, where each $g_i$ guides a different conditionally independent property.
>
> ---
>
> ## 2. Experiment with non-differentiable potential
>
> Please see our common response (item 2).
>
> ---
>
> ## 3. Real data with disconnected modes
>
> The materials generation task is precisely such a realistic example with disconnected modes. Because atom types are discrete, the target space contains large disconnected modes corresponding to distinct atomic compositions. We believe Table 2 reflects D-Flow's difficulty navigating these disconnected modes - similar to the toy example in Figure 2, but in a high-dimensional, realistic setting.
>
> ---
>
> ## 4. Image guidance tasks
>
> ESS-Flow could perform well at tasks with less informative conditioning, such as class-conditional image generation. However, please see our reason for not focusing on vision tasks in the common response (item 5).
>
> ---
>
> ## 5. Usage of CHGNet
>
> We use CHGNet for materials relaxation and energy prediction following prior work in FlowMM. While newer models like MACE, eSEN, or UMA may provide more accurate energy estimates, our focus is on comparing relative sample quality rather than absolute accuracy. So we will report quality metrics for all methods, including an extended (S.U.N.T) metric measuring the proportion of samples that are stable, unique, novel, and meet the target property threshold. CHGNet provides a consistent framework for this comparison, and we expect the relative performance rankings would remain similar with different energy models.

---

> > ### Author Response · Authors · 2025-12-01
> > **Follow-up**
> >
> > We have now updated the paper with additional experiments and details to address the main concerns.
> >
> > ---
> >
> > ## 2. Experiment with non-differentiable potential
> >
> > We have updated the material generation experiments in Section 5.1 by including material generation with target space-group symmetry. For this task, the target property is discrete (one of 230 possible space groups), and the potential function is a binary indicator computed using the non-differentiable external program (spglib), making gradient-based methods fundamentally inapplicable. ESS-Flow successfully generates 92.3\% of samples with the target P$6_3$/mmc space group, compared to only 2.5\% when sampling unconditionally from the prior.
> >
> > ---
> >
> > ## 5. Usage of CHGNet
> >
> > We have updated Table 3 to include S.U.N.T. metrics (stability, uniqueness, novelty, and threshold) for all methods considered, enabling direct comparison. ESS-Flow achieves significantly better S.U.N.T. rates than all baselines across all tasks. We expect the relative performance rankings would remain the same with different energy models to predict the stability rate.

---

### Official Review · Reviewer_sKKR · 2025-10-29

**Soundness:** 3
**Presentation:** 3
**Contribution:** 3
**Rating:** 4
**Confidence:** 3

**Summary:**

This paper proposed a plug-and-play controllable generation method for flow-based generative models, which is based on sampling the controlled prior distribution through elliptical sliced sampling (ESS), and then run the ODE to convert the prior samples to samples from the tilted target distribution. The method avoids the need to compute the Jacobian of the flow map $T _ \theta$, and does not require the tilt to be differentiable. The authors further proposed to use a coarse discretization of the flow ODE as a proposal for the transition kernel and reweight the samples with the evaluation of the tilt on the final samples to reduce the computational cost.

**Strengths:**

Unlike most existing controllable generation methods such as CFG and DPS that involve the computation of an extra term in the SDE (often requires approximation), the proposed method just treat the flow ODE as a black-box and only modifies the prior distribution, and instead of doing gradient-based optimization, it directly samples from the controlled prior distribution. This makes the method simple and concise, and I personally appreciate this idea.

The paper also provided comparisons of the proposed method with existing controllable generation methods based on optimization. I'm not familiar in the domain of material design and protein structure prediction, but the experimental results look nice.

**Weaknesses:**

The authors provided limited background information about the ESS algorithm, which I believe most of the readers in the ML community should not be familiar with. We know that there are lots of zeroth-order sampling methods for sampling from $\pi(z)\propto p(z) g(T _ \theta(z))$, such as rejection sampling, the MH algorithm, and proximal sampler (https://proceedings.mlr.press/v134/lee21a.html). What's the insight behind ESS, and why is the Gaussian prior important for it to work? (Also I'm not quite in favor of the use of the abbreviation ESS in this paper, since it is also used to refer to "effective sample size" in the literature of Monte Carlo methods, which may cause some misunderstanding. But it's fine if you keep it.)

For the experiments, I think it would be more convincing if the authors can also consider some image generation tasks, which are more familiar to most ML researchers, and can better demonstrate the effectiveness of the proposed method. Also, as the authors have introduced the multi-fidelity version of the proposed method through a coarse discretization of the flow ODE, it would be interesting to see some ablation studies on the choice of the discretization step size and its impact on the generation quality and computational cost.

I'm happy to raise the score if these issues can be addressed during rebuttal.

**Questions:**

1. What's the typical number of rejections needed in one iteration of ESS in order to get an accepted sample?

2. We know that diffusion model and flow matching allow one-step prediction of $\mathbb{E}[x _ 1|x _ t]$. For faster sampling, instead of still doing discretization of the flow ODE, do you think it is possible to replace the flow map $T _ \theta$ with a one or few step predictor in the proposed method? If a consistency model is available, we can even directly predict the final sample from the prior sample in one step, which may further reduce the computational cost.

3. The main text of the paper assumes the flow model is trained for Euclidean data, but in the experiments it turns out that the whole framework can also be applied to manifold data as long as we have a flow-based generative model, which I believe is an advantage of the proposed method, and should be highlighted more in the paper. I'm not quite familiar with the literature of manifold, but is there anything that we need to pay attention to when applying the proposed method to manifold data? For example, do we need to modify the ESS algorithm in any way?

4. In table 2 and 3, it would be better to highlight the best results in bold font for better readability.

---

> ### Author Response · Authors · 2025-11-22
> **Response to Reviewer sKKR**
>
> We thank the reviewer for their thoughtful feedback. We believe the clarifications and additional experiments below address the main concerns raised.
>
> ---
>
> ## 1. Elliptical slice sampling
>
> We appreciate the opportunity to clarify the sampling method and will expand its description in the revision. In ESS, proposals are drawn along an ellipse passing through the current state and a random sample from the prior. For Gaussian priors, proposals are also distributed as per the prior. Unlike Metropolis-Hastings, where a rejected proposal requires drawing from a new ellipse, ESS adaptively shrinks the current ellipse and continues sampling until acceptance. This mechanism can be seen as an adaptive step-size that enables larger jumps without tuning. Prior work has shown the statistical efficiency (in terms of effective sample size) of ESS to be dimension-independent (Natarovskii et al., 2021), with extensions to infinite-dimensional Hilbert spaces being well-defined (Hasenpflug et al., 2025). We note that ESS is a well-established terminology for elliptical slice sampling in the MCMC literature (Hasenpflug et al., 2025; Bitterlich et al., 2025; Nishihara et al., 2014).
>
> ---
>
> ## 2. Experiments on image restoration
>
> Please see our common response (item 5).
>
> ---
>
> ## 3. Sampling efficiency of ESS-Flow
>
> Please see our common response (item 4).
>
> ---
>
> ## 4. Faster generative models
>
> Thank you for the suggestion! Indeed, methods that speed up generation in flow-based models, such as consistency models or flow-matching with minibatch OT coupling, can be applied to ESS-Flow as is, to significantly reduce computational cost. We will clarify the generality of ESS-Flow with respect to the choice of generative prior model in the revised manuscript.
>
> ---
>
> ## 5. Manifold data
>
> This is an excellent point that we will clarify in the revised manuscript. ESS-Flow only requires the latent/source distribution to be Gaussian in $\mathbb{R}^d$ and a deterministic transport map between source and target. Because the transport map is used as a black box for point-wise evaluations only, ESS-Flow works with models generating manifold data (such as FlowMM used in our materials experiments, which is trained with Riemannian flow matching) without any modifications.
>
> ---
>
> **References**
> 1. Natarovskii et al., 2021, Geometric convergence of elliptical slice sampling, ICML 2021
> 2. Hasenpflug et al., 2025, Reversibility of elliptical slice sampling revisited, Bernoulli, 31(2)
> 3. Bitterlich et al., 2025, Delayed Acceptance Elliptical Slice Sampling, PAMM, 25(1)
> 4. Nishihara et al., 2014, Parallel MCMC with generalized elliptical slice sampling, JMLR 2014

---

> > ### Author Response · Authors · 2025-12-01
> > **Follow-up**
> >
> > We have now updated the paper with additional experiments and details to address the main concerns.
> >
> > ---
> >
> > ## 1. Elliptical slice sampling (ESS)
> >
> > We have expanded our description of ESS in Section 4.1 to provide more intuition for readers unfamiliar with the method.
> >
> > ---
> >
> > ## 3. Sampling efficiency
> >
> > We now report average number of evaluations per MCMC step ($n_\text{ellipse}$) for ESS-Flow in Tables 5, 7. Values range from 4.8-17.4 for materials tasks and 2.7-4.1 for GMM experiments.
> >
> > ---
> >
> > **Multi-fidelity ESS-Flow**: We have included a detailed comparison between standard ESS-Flow samples and the importance-weighted multi-fidelity version in Appendix A.3. While the multi-fidelity approach with importance reweighting can slightly improve accuracy for tasks with moderate target concentration, it performs poorly for sharply concentrated targets. The uniqueness rates drop significantly due to resampling from the discrete weighted distribution.

---

### Official Review · Reviewer_iqsg · 2025-10-31

**Soundness:** 2
**Presentation:** 3
**Contribution:** 2
**Rating:** 2
**Confidence:** 3

**Summary:**

The paper introduces ESS-Flow, a training-free and gradient-free guidance method for flow-based generative models. The key idea is to do Monte Carlo inference directly in the source space, where the prior is Gaussian, by sampling from a distribution of the form $\pi(z) \propto g(T_\theta(z))p(z)$. This makes the method applicable to quantized/materials settings and to non-differentiable observation or reward functions. The authors further propose a multi-fidelity variant where the authors sample using a coarse ODE discretization and reweight using a fine discretization to reduce computation. Experiments on materials and protein structure prediction show that ESS-Flow achieves nearly SOTA results on all metrics for materials and comparable performance on protein generation metrics.

**Strengths:**

1. The paper is clearly written, with a coherent structure and well-motivated.

2. The proposed method is simple to implement and achieves strong results on the material generation task.

**Weaknesses:**

1. The paper lacks experiments on standard image-domain tasks (e.g. inpainting, deblurring), which are commonly used in related work such as D-Flow and PnP-Flow.


2. While the empirical contribution is solid, the theoretical contribution is relatively modest.

3. One of the main points of the contribution is having this apply to non-differentiable rewards/potentials yet there are no experiments demonstrating this capability.

**Questions:**

1. What are the quality metrics in Table 3 for the competing methods? It would be helpful to report the same set of metrics so we can compare ESS-Flow directly to the baselines.

2. How does multi-fidelity ESS compare to standard ESS on the metrics reported in Table 2 and Table 4.

3. What are the acceptance rates of the sampling?

---

> ### Author Response · Authors · 2025-11-22
> **Response to Reviewer iqsg**
>
> We thank the reviewer for their feedback. We believe several of raised concerns can be addressed through additional experiments and clarifications. Below we respond to the main points.
>
> ---
>
> ## 1. Experiments on image restoration [W1]
>
> Please see our common response (item 5).
>
> ---
>
> ## 2. Theoretical contributions [W2]
>
> We acknowledge that our work builds on established MCMC theory rather than developing new theoretical results. However, we view the principled application of ESS to flow models as a meaningful contribution. Many existing controlled generation methods lack theoretical guarantees and may not target the correct posterior distribution, whereas ESS-Flow is provably consistent and preserves the theoretical properties of ESS.
>
> ---
>
> ## 3. Experiment with non-differentiable potential [W3]
>
> Please see our common response (item 2).
>
> ---
>
> ## 4. Metrics [Q1, Q2]
>
> We originally reported quality metrics in Table 3 only for ESS-Flow because other methods frequently failed to produce samples meeting target properties. However, we agree that direct comparison would be more informative. We will report quality metrics for all methods, including an extended (S.U.N.T) metric measuring the proportion of generated samples that are stable, unique, novel, and within the target property threshold. We will also report metrics comparing the performance of multi-fidelity ESS versus standard ESS.
>
> ---
>
> ## 5. Acceptance rate [Q3]
>
> Please see our common response (item 4).

---

> > ### Author Response · Authors · 2025-12-01
> > **Follow-up**
> >
> > We have now updated the paper with additional experiments and details to address the main concerns.
> >
> > ---
> >
> > ## 3. Experiments with non-differentiable potential [W3]
> >
> > We have updated the material generation experiments in Section 5.1 by including material generation with target space-group symmetry. For this task, the potential function is a binary indicator computed using the non-differentiable external program spglib, making gradient-based methods fundamentally inapplicable. ESS-Flow successfully generates 92.3\% of samples with the target P$6_3$/mmc space group, compared to only 2.5\% when sampling unconditionally from the prior.
> >
> > ---
> >
> > ## 4. Metrics [Q1, Q2]
> >
> > We have updated Table 3 to include S.U.N.T. rates for all baseline methods, enabling direct comparison with ESS-Flow. ESS-Flow achieves the highest S.U.N.T. rate across all tasks considered. We have also included additional comparison between standard ESS-Flow samples and the importance weighted multi-fidelity version in Appendix A.3.
> >
> > ---
> >
> > ## 5. Acceptance rate [Q3]
> >
> > ESS has an acceptance probability of 1 for the MCMC chain (no repeated states), but the bracket shrinkage procedure requires multiple evaluations along the ellipse. We now report average number of evaluations per MCMC step ($n_\text{ellipse}$) for ESS-Flow in Tables 5, 7. Values range from 4.8-17.4 for materials tasks and 2.7-4.1 for GMM experiments.

---

### Official Review · Reviewer_24zR · 2025-11-03

**Soundness:** 2
**Presentation:** 3
**Contribution:** 3
**Rating:** 6
**Confidence:** 3

**Summary:**

The authors introduce ESS-Flow, a gradient-free method for controlled generation in the setting of generative modelling with flow matching models. The authors perfoming Bayesian inference in source space using Elliptical Slice Sampling, which enables conditional generating without requiring gradients. The authors demonstrate their approach on various applications ranging from materials to proteins.

**Strengths:**

[S1] The gradient-free nature of this approach is appealing. There have been other works for source space sampling such as , but this still required gradients, wheras this approach here circumvents this via the Jacobian cancellation and the ESS approach.

[S2] One common with gradient-based optimisers in diffusion samplers are the multitude of often brittle hyperparameters like guidance scales or other schedulers; this approach here seems to be less reliant on these.

[S3] Good motivation of the different components introduced with formal justifications.

**Weaknesses:**

[W1] The authors demonstratet that they can avoid the Jacobian computation, but ESS-Flow still requires many evals (>1000 MCMC steps) of the transport map, hurting the efficiency of the approach

[W2] The authors openly describe the limitation of ESS-Flow in cases where the target is constrained on a lower-dim manifold, but claim that in scientific domains the target distribution is not overly collapsed. However, in many applications like in protein design the target distribution lies exactly on such a lower dim manifold with most of the target space being invalid sampels; some more explanation why the authors think that this is not the case in many scientific applciations would help here.

[W3] In many scientific applications, people have circumvented the non-differentiability of categorical sequences via soft relaxations similar to the atom relaxation the authors use for their comparisons. In approaches like BindCraft (Pacesa et al, 2025 Nature), this works remarkably well, so the authors should potentially try to tune that baseline to see if it as strong as it can be.

[W4] While some of the baselines in the protein structure prediction case of Figure 4 look unrealistic, ESS-Flow also seems to have biophysical implausibilities, and the ELBO of the model only partially captures these things. A more fundamental evaluation like counts of clashes could demonstrate how good the structures actually are; the RMSD values above 10 suggest that all baselines seem pretty far off.

**Questions:**

[Q1] The case studies all have quite low dimensionality, how does the approach scale to high dimensional problems?

[Q2] Given the authors say their method does not work well with priors that poorly inform the target distribution, can this statement be made more exact? ie is there a quantity that one can look at to see if the approach will work or not?

---

> ### Author Response · Authors · 2025-11-22
> **Response to Reviewer 24zR**
>
> We thank the reviewer for their positive assessment of our work and thoughtful feedback. Below we address the main concerns regarding efficiency, applicability, and evaluation.
>
> ---
>
> ## 1. Runtime cost of ESS-Flow [W1]
>
> Please see our common response (item 3).
>
> ---
>
> ## 2. Limitations of ESS-Flow [W2, Q2]
>
> We appreciate the opportunity to clarify this important point. While much of the data space $\mathcal{X}$ often corresponds to invalid samples, the limitation we refer to is when the target distribution in the _source_ space $\pi(z) \propto g(T_\theta(z)) p(z)$ is highly concentrated. Since the prior in $\mathcal{Z}$ is standard Gaussian, this only occurs when the pulled-back potential $g(T_\theta(z))$ is overly collapsed - as in inverse problems like image deblurring where observations are highly informative and typically only the MAP solution is of interest. In contrast, many scientific applications like drug or materials design can have multiple candidate solutions for a specific objective, making posterior sampling methods like ESS-Flow well-suited for exploring diverse solutions rather than finding a single optimum. Clearly, this is not universally true for applications in science, but our claim is that there _exists a range of important problems in the scientific domain where this is indeed the case_. Our empirical results on guided materials generation, for instance, illustrate this point. On the contrary, we believe that most problems in computer vision (or more specifically, image restoration) typically have highly concentrated pull-back potentials, but such problems are not our main focus.
>
> While we don't have a single diagnostic quantity to predict applicability in advance, analyzing the potential function can provide guidance. If the potential is highly concentrated (e.g., if it corresponds to the likelihood of a high-dimensional and informative observation), then ESS-Flow might not be the most suitable method. On the other hand, if the user believes that there is a diversity of samples that all "satisfy" the potential (e.g. a diversity of materials/molecules with a particular sought after property), then ESS-Flow can be expected to perform well.
>
> ---
>
> ## 3. Limitations of soft relaxations [W3]
>
> We agree that soft relaxations of categorical variables can enable gradient-based methods in many cases. However, ESS-Flow addresses scenarios, where the potential or likelihood function $g(x)$ itself is non-differentiable, even after such relaxations. For example, in cases where $g(x)$ is computed by an external simulator or involves discrete operations, gradient-based methods remain inapplicable regardless of whether the inputs are relaxed. To demonstrate ESS-Flow's capability in this setting, we will include experiments on materials generation with discrete target space-group symmetry, where the likelihood is computed as a binary indicator using a non-differentiable external program.
>
> ---
>
> ## 4. Realism of protein structures [W4]
>
> We agree that ELBO alone is insufficient for assessing structural realism, and we will include additional metrics such as clash counts in the revision. Our primary aim with this experiment was to demonstrate that methods operating in target space are more susceptible to generating unrealistic samples by deviating from the generative model prior, compared to source space methods. We acknowledge that this particular task remains challenging for all methods considered, with significant room for improvement.
>
> ---
>
> ## 5. Scaling with dimensions [Q1]
>
> Please see our common response (item 1).

---

> > ### Author Response · Authors · 2025-12-01
> > **Follow-up**
> >
> > We have now updated the paper with additional experiments and details to address the main concerns.
> >
> > ---
> >
> > ## 1. Runtime cost of ESS-Flow [W1]
> >
> > We have added Appendix A.2 with detailed runtime cost analysis showing function evaluation counts for all methods, and peak GPU memory usage across tasks. The low memory requirement allows ESS-Flow to use much larger batch sizes, which can partially offset the higher function evaluation cost in practice. We have also updated the results in Section 5.1 by running ESS-Flow with 1000 parallel MCMC chains for 200 steps (500 for the band gap task), demonstrating that competitive performance can be achieved with reduced sampling per chain.
> >
> > ---
> >
> > ## 3. Limitations of soft relaxations [W3]
> >
> > We have added an experiment to generate materials with target space-group symmetry in Section 5.1 where the potential is a binary indicator computed using a non-differentiable external program (spglib). This makes gradient-based methods fundamentally inapplicable regardless of whether the atomic representation is relaxed. ESS-Flow successfully generates 92.3\% of samples with the target P$6_3$/mmc space group, compared to only 2.5\% when sampling unconditionally from the prior.
> >
> > ---
> >
> > ## 4. Realism of protein structure [W4]
> >
> > We have included clash counts for the protein structures generated by each method in Table 4. The results show that ADP-3D and DAPS produce structures with much higher clashes, compared to D-Flow and ESS-Flow. In agreement with the ELBO, this demonstrates that methods operating in the source space rather than optimizing directly in the target space produce comparably better quality samples. However, we acknowledge that this task remains challenging for all methods considered, including ESS-Flow, as indicated by the high RMSD values.
> >
> > ---
> >
> > ## 5. Scaling with dimensions [Q1]
> >
> > We have added Appendix A.1 with empirical evaluation on Gaussian  mixture models with linear-Gaussian observations across varying dimensions $(d_x \in {8, 80, 800}, d_y \in {1, 2, 4})$. Results confirm that ESS-Flow maintains stable sampling efficiency and low Sliced Wasserstein Distance to samples from the exact posterior as dimensionality increases. Please see our common response (item 1) for detailed discussion.

---

### Official Review · Reviewer_DpS7 · 2025-11-03

**Soundness:** 3
**Presentation:** 3
**Contribution:** 3
**Rating:** 6
**Confidence:** 3

**Summary:**

The paper introduces ESS-Flow, a training-free and gradient-free approach for controlled generation using pretrained diffusion or flow-based models. The method reframes inference in the Gaussian latent source space and applies elliptical slice sampling (ESS) instead of gradient-based updates, using a change of variables to apply updates in the data space. The main goal is to enable preference alignment when gradients are unavailable or unreliable, such as in cases involving quantized or simulator-based objectives.

**Strengths:**

- The paper introduces a new and practically useful application of elliptical slice sampling to flow- and diffusion-based generative models. While ESS itself is a known MCMC method, its use within the latent Gaussian space of pretrained flow-based models is novel and interesting.
- The algorithm is designed such that jacobian determinants of transport maps don’t need to be computed, enabling a scalable and efficient algorithm.


- The theory is sound, and the algorithm preserves theoretical guarantees from the original ESS method, making interpretability for downstream researchers easier.
Reported results compared to baselines show a non-negligible improvement in matching a new target energy function.


- Overall, the paper connects ideas from generative modeling, Bayesian inference, and MCMC in a coherent and insightful way. It is an interesting paper that I believe would bring a net positive to the research community, which would be strengthened by addressing the weaknesses below.

**Weaknesses:**

- While 0th-order methods can benefit greatly in settings with unreliable gradients, they can also face severe scaling issues in high-dimensional spaces. Having an experiment showing performance as problem dimension scales would help further inform future readers when they should use ESS-flow vs. a gradient-based method.


- While results on the provided experiments show that the 0th order methods are outperforming the gradient-based methods, none of the experimental settings to my understanding actually fall under the non-differentiable setting that the paper proposes to address. It would help to either (a) include a relevant experiment setting where gradient information is truly intractable to retrieve, or (b) demonstrate that in the reported settings, the gradient structure is highly unideal for gradient-based methods, e.g. high Lipschitz constant (also, for line 290, maybe better to call them gradient-based rather than optimization-based, since there are many 0th order optimization algorithms).
More statistical details on the experimental setup are needed, e.g. some std’s in table 2 are higher in magnitude than the mean.


- There are some missing baselines/ablations (e.g. adjoint matching [Domingo-Enrich et al.], non-ESS-based source space MCMC) that would help clarify whether the main benefit comes from being gradient-free or from the specific ESS mechanism.

**Questions:**

- How does ESS-Flow scale with increasing latent dimension? Does the acceptance rate or effective sample size drop significantly in higher dimensions?


- What is the runtime cost compared to the baselines?

---

> ### Author Response · Authors · 2025-11-22
> **Response to Reviewer DpS7**
>
> We thank the reviewer for their constructive feedback and positive assessment of our work. Below we address the main concerns regarding scalability, experimental settings, and baselines.
>
> ---
>
> ## 1. Scaling with dimensions
>
> Please see our common response (item 1).
>
> ---
>
> ## 2. Experiment with non-differentiable potential
>
> Please see our common response (item 2).
>
> ---
>
> ## 3. Other baselines to  clarify whether the main benefit comes from being gradient-free or from the specific ESS method
>
> Thank you for the comment. We would argue that the main benefit comes from both being gradient-free and targeting the correct posterior - our choice of ESS is motivated by its proven efficiency for such problems, particularly in high dimensions (see item 1 in our common response). To the best of our knowledge, there are no existing methods that target the source space posterior for flow-based models using _gradient-free_ sampling. Prior methods such as Purohit et al., 2025 use Langevin Monte Carlo and Graham et al., 2017 use Hamiltonian Monte Carlo.
>
> We appreciate the suggestion to also compare against finetuning-based methods like adjoint matching. However, our work focuses specifically on training-free methods that use the pretrained models without modification.
>
> ---
>
> ## 4. Runtime cost of ESS-Flow
>
> Please see our common response (item 3).
>
> ---
>
> **References**
> 1. Purohit et al., 2025, Consistency Posterior Sampling for Diverse Image Synthesis, CVPR 2025
> 2. Graham et al., 2017, Asymptotically exact inference in differentiable generative models, AISTATS 2017

---

> > ### Author Response · Authors · 2025-12-01
> > **Follow-up**
> >
> > We have now updated the paper with additional experiments and details to address the main concerns.
> >
> > ---
> >
> > ## 1. Scaling with dimensions
> >
> > We have added Appendix A.1 with empirical evaluation on Gaussian  mixture models with linear-Gaussian observations across varying dimensions $(d_x \in {8, 80, 800}, d_y \in {1, 2, 4})$. Results confirm that ESS-Flow maintains stable sampling efficiency and low Sliced Wasserstein Distance to samples from the exact posterior as dimensionality increases. Please see our common response (item 1) for detailed discussion.
> >
> > ---
> >
> > ## 2. Experiment with non-differentiable potential
> >
> > We have added an experiment to generate materials with target space-group symmetry in Section 5.1 where the potential is a binary indicator computed using a non-differentiable external program (spglib). This makes gradient-based methods fundamentally inapplicable. ESS-Flow successfully generates 92.3\% of samples with the target P$6_3$/mmc space group, compared to only 2.5\% when sampling unconditionally from the prior.
> >
> > ---
> >
> > ## 4. Runtime cost of ESS-Flow
> >
> > We have added Appendix A.2 with detailed runtime cost analysis showing function evaluation counts for all methods, and peak GPU memory usage across tasks. The low memory requirement allows ESS-Flow to use much larger batch sizes, which can partially offset the higher function evaluation cost in practice.
> >
> > ---
> >
> > **Statistical details**: We have improved the experimental evaluation by increasing the sample size to 1000 for all materials tasks and updated the results (Tables 2-3, Figure 3). ESS-Flow now uses 1000 parallel MCMC chains to reduce autocorrelation and improve diversity.

---

### Author Response · Authors · 2025-11-22
**Common Response**

We thank all the reviewers for their thoughtful feedback. Below we address common concerns raised across multiple reviews.

---

## 1. Scaling with dimensions

Elliptical slice sampling has strong theoretical properties in high dimensions, with prior works showing its statistical efficiency is dimension-independent (Natarovskii et al., 2021), and extensions to infinite dimensional Hilbert space is well-defined (Hasenpflug et al., 2025). To demonstrate this for ESS-Flow, we are currently working on an empirical evaluation on simulated data to test how the method scales with dimension, and will update you on the results as soon as they are available.

---

## 2. Experiment with non-differentiable potential

Our reason for choosing differentiable potentials in the evaluation was to facilitate comparisons with gradient-based methods. However, we fully agree that demonstrating performance in truly non-differentiable settings would strengthen our claims. We will include experiments on materials generation with discrete target space-group symmetry, where the likelihood is computed as a binary indicator using a non-differentiable external program, making gradient-based methods inapplicable for this task.

---

## 3. Runtime cost of ESS-Flow

ESS-Flow requires comparable or more function evaluations than gradient-based baselines. However, it has significantly lower memory requirements. For example, the memory cost of standard gradient propagation through an ODE is linear in the number of discretization steps. ESS-Flow thus allows us to use much larger batch sizes that can partially offset the overall runtime cost. We will include a detailed breakdown of the compute and memory requirements for the different methods in the revised manuscript.

---

## 4. Sampling efficiency of ESS-Flow

In elliptical slice sampling, proposals are drawn along a shrinking ellipse until acceptance, with bracket shrinkage at each rejection. Hence, the MCMC sample path itself has an acceptance probability of 1 (i.e., there are no "repeated states" in the chain, as one would expect for e.g. Metropolis--Hastings sampling). However, the shrinkage procedure requires evaluating the target multiple times along the ellipse. For our tasks, this typically requires 6-12 evaluations per accepted sample. We will report the average number of function evaluations per MCMC step in the revised manuscript to provide a clearer picture of sampling efficiency.

---

## 5. Experiments with image restoration

We appreciate this suggestion but respectfully maintain our focus on scientific applications where existing methods face challenges. ESS-Flow is not intended to be used on image restoration problems like deblurring, and we report this as a limitation in the manuscript (see reply to Reviewer 24zR for a more in-depth explanation of why this is the case). These tasks are well suited for gradient-based methods - observations are highly informative and gradients are readily available - and, indeed, there exists a plethora of methods that successfully address image restoration. In contrast, for the class of problems that is our primary focus, such as material generation, gradient-based methods often require approximations and substantial tuning while ESS-Flow works robustly out-of-the-box. This reflects our goal to design for settings where diverse posterior samples are needed and gradients are unreliable. We believe the community benefits from methods with well-characterized strengths and use-cases, rather than claiming universal applicability.

---

**References**
1. Natarovskii et al., 2021, Geometric convergence of elliptical slice sampling, ICML 2021
2. Hasenpflug et al., 2025, Reversibility of elliptical slice sampling revisited, Bernoulli, 31(2)

---

> ### Author Response · Authors · 2025-12-01
> **Summary of Revisions**
>
> We have carefully addressed the concerns raised and made substantial improvements to the manuscript. Below we summarize the main changes.
>
> ---
>
> ## 1. Scaling with dimensions
>
> We have added Appendix A.1 with empirical evaluation on Gaussian  mixture models with linear-Gaussian observations across varying dimensions $(d_x \in {8, 80, 800}, d_y \in {1, 2, 4})$. Results confirm that ESS-Flow maintains stable sampling efficiency and low Sliced Wasserstein Distance to samples from the exact posterior as dimensionality increases.
>
> ---
>
> ## 2. Non-differentiable potentials
>
> We have added an experiment to generate materials with target space-group symmetry in Section 5.1 where the potential is a binary indicator computed using a non-differentiable external program (spglib). ESS-Flow successfully generates 92.3\% of samples with the target P$6_3$/mmc space group compared to only 2.5\% from unconditional sampling, demonstrating its effectiveness in this challenging setting.
>
> ---
>
> ## 3. Runtime cost
>
> We have added Appendix A.2 with detailed runtime cost analysis showing function evaluation counts for all methods, and peak GPU memory usage across tasks. The low memory requirement allows ESS-Flow to use much larger batch sizes, which can partially offset the higher function evaluation cost in practice.
>
> ---
>
> ## 4. Sampling efficiency
>
> We now report average number of evaluations per MCMC step ($n_\text{ellipse}$) for ESS-Flow in Tables 5, 7. Values range from 4.8-17.4 for materials tasks and 2.7-4.1 for GMM experiments.
>
> ---
>
> Additionally, we have made the following improvements.
>
> - **Material generation**: We have updated all the results (Tables 2-3, Figure 3) in Section 4.1 with evaluations over 1000 generated samples. ESS-Flow now uses 1000 parallel MCMC chains to reduce sample autocorrelation and improve diversity. And we report the S.U.N.T. rate for all baselines (Table 3), enabling direct quantitative comparison. Results show ESS-Flow achieves highest S.U.N.T. rates across all tasks.
>
> - **Multi-fidelity evaluation**: We have revised the effective sample sizes in Section 5.1.1, and added Appendix A.3 with additional evaluations comparing standard ESS-Flow against the importance reweighted samples.
>
> - **Protein structure evaluation**: We have added clash counts (Table 4) as a direct measure of structural realism for the generated proteins

---

### Meta-Review · Area_Chair_Nmhn · 2026-01-08

**Summary:**

This paper introduces ESS-Flow, a method for guiding flow-based models using Elliptical Slice Sampling (ESS) in the latent source space. The core promise is "gradient-free" guidance, enabling control via non-differentiable or black-box potentials. The initial reviews were mixed (scores 2, 4, 6, 6, 6). The primary critique was a "validity gap": the paper claimed to solve non-differentiable problems but only benchmarked on differentiable ones (where gradient methods are superior). Secondary concerns focused on the computational inefficiency of MCMC sampling in high dimensions and the omission of standard image restoration benchmarks.

**Reviewer Concerns:**

**Addressed**:
- Proof of Concept for Non-Differentiable Potentials (DpS7, iqsg, C4du): The reviewers rightly pointed out that without a non-differentiable experiment, the method had no demonstrated advantage over gradient-based baselines. The authors added a material generation task where the reward (space-group symmetry) is computed by an external black-box simulator (spglib). The method achieved a 92% success rate versus 2.5% for unconditional sampling. This provides the factual evidence that the method works where gradient methods fail.

**Partially Addressed**:
- Scalability to High Dimensions (DpS7, 24zR): The reviewers asked how ESS scales with dimensionality. The authors provided a synthetic GMM experiment. While this shows stability in principle, a toy GMM does not empirically prove scalability for the complex, high-dimensional scientific inverse problems the paper targets.
- Sampling Efficiency (sKKR, 24zR): Reviewers noted the high cost of MCMC. The authors provided a runtime analysis confirming that ESS-Flow requires significantly more function evaluations than gradient methods (4-17x per step). They argue that lower memory usage allows for larger batch sizes to compensate, but this confirms that the method is computationally inefficient per sample.

**Outstanding**:
- General Applicability / Image Benchmarks (iqsg, sKKR, C4du): Multiple reviewers requested standard image restoration tasks (inpainting, deblurring) to place the method in the broader literature. The authors refused, arguing that ESS fails on "concentrated posteriors" typical of imaging. While scientifically honest, this refusal explicitly defines the method as a niche tool for "broad posterior" scientific problems, rather than a general-purpose guidance algorithm.

**Reviewer Scores:**

- Reviewer DpS7 (Original: 6): Predicted: 6. Their primary request (non-differentiable experiment) was met with new experiment.
- Reviewer 24zR (Original: 6): Predicted: 6 . While they received the requested data, the results confirmed the high computational cost and "toy" nature of the scaling experiment. A score increase is not guaranteed.
- Reviewer iqsg (Original: 2): Predicted: 4. The low rating was heavily rely on the lack of non-differentiable experiments. Since this gap was filled, the author may increase the score, though the lack of image benchmarks limits enthusiasm.
- Reviewer sKKR (Original: 4): Predicted: 4. The reviewer indicates "happy to raise the score" if issues were addressed. The efficiency data and ESS background were provided, though the refusal to run image tasks may limit the increase. This reviewer disclaimed that their expertise is not in protein or AI4science.
- Reviewer C4du (Original: 6): Predicted: 6. The discrete property guidance experiment directly answered their main request.

---

### Decision · Program_Chairs · 2026-01-26

Reject